# Effective Approximation Method for Nanogratings-induced Near-Field Radiative Heat Transfer

**DOI:** 10.3390/ma15030998

**Published:** 2022-01-27

**Authors:** Yang Liu, Fangqi Chen, Andrew Caratenuto, Yanpei Tian, Xiaojie Liu, Yitong Zhao, Yi Zheng

**Affiliations:** 1Department of Mechanical and Industrial Engineering, Northeastern University, Boston, MA 02115, USA; liu.yang18@northeastern.edu (Y.L.); chen.fangq@northeastern.edu (F.C.); caratenuto.a@northeastern.edu (A.C.); tian.yan@northeastern.edu (Y.T.); liu.xiaojie@northeastern.edu (X.L.); 2Department of Mechanical Engineering, California State Polytechnic University Pomona, Pomona, CA 91768, USA; YitongZhao@cpp.edu

**Keywords:** near-field radiative heat transfer, effective approximation NFRHT method, effective medium theory, nanostructures

## Abstract

Nanoscale radiative thermal transport between a pair of metamaterial gratings is studied within this work. The effective medium theory (EMT), a traditional method to calculate the near-field radiative heat transfer (NFRHT) between nanograting structures, does not account for the surface pattern effects of nanostructures. Here, we introduce the effective approximation NFRHT method that considers the effects of surface patterns on the NFRHT. Meanwhile, we calculate the heat flux between a pair of silica (SiO_2_) nanogratings with various separation distances, lateral displacements, and grating heights with respect to one another. Numerical calculations show that when compared with the EMT method, here the effective approximation method is more suitable for analyzing the NFRHT between a pair of relatively displaced nanogratings. Furthermore, it is demonstrated that compared with the result based on the EMT method, it is possible to realize an inverse heat flux trend with respect to the nanograting height between nanogratings without modifying the vacuum gap calculated by this effective approximation NFRHT method, which verifies that the NFRHT between the side faces of gratings greatly affects the NFRHT between a pair of nanogratings. By taking advantage of this effective approximation NFRHT method, the NFRHT in complex micro/nano-electromechanical devices can be accurately predicted and analyzed.

## 1. Introduction

When the distance between two objects is comparable to or less than the thermal wavelength, the photon tunneling effect plays an essential role in the thermal radiative transfer, greatly enhancing the near-field radiative heat transfer (NFRHT) past the Planckian blackbody limit by several orders of magnitude [1,2,3,4,5,6]. Numerous theoretical studies on the NFRHT between various materials and nanostructures have been performed, such as transfer between two planar surfaces [1,7,8], two nanoparticles [8,9], two gratings [10,11,12,13,14], one sphere and plane [15], three bodies [16], and so on [17,18,19,20,21,22,23,24,25,26,27,28,29,30,31,32]. Recently, many experimental reports have indicated that the NFRHT can exceed the blackbody limit for a plane–plane configuration [33,34,35,36,37,38]. Therefore, NFRHT is of great significance in a variety of engineering applications, such as energy management [17,39,40,41,42], sensing, and micro/nano-electromechanical systems (M/NEMS) [43,44,45,46].

To theoretically calculate NFRHT, there are two types of fundamental frameworks. The first one is the fluctuational electrodynamics (FE) formalism by combining the Maxwell equations with the fluctuation–dissipation theorem (FDT), specifically including analytical formalism [47], the scattering matrix method [48], thermal discrete dipole approximation (TDDA) [49], fluctuating surface-current approach (FSC) [50], and the finite difference time domain method (FDTD). The second one is based on the kinetic theory (KT) approach combined with the Boltzmann transport equation (BTE) to solve the distribution function of thermal photons [51,52].

In the domain of radiative heat transfer, nanogratings deserve special attention. The NFRHT analyses between nanogratings typically use the effective medium theory (EMT) to treat the nanograting structures as homogeneous films and approximate them as systems of infinite size. When the vacuum distance between a pair of nanogratings is less than the grating period, EMT-based methods are not applicable anymore [53,54,55,56,57,58]. In addition, the surface pattern effects of gratings (e.g., the lateral displacement of nanogratings) also pose challenges when using the EMT to investigate the NFRHT between relatively displaced nanogratings. The analysis of a finite-sized system that considers NFRHT effects beyond the EMT approximation for a pair of relatively displaced nanogratings is strongly desired. Liu et al. [14,59] assessed the accuracy of the EMT approximation applied to NFRHT between two nanogratings by using the scattering theory method. Although the scattering matrix method is specially suited to deal with layered planar structures, including periodically patterned planar systems such as nanogratings or photonic crystals, this method requires tremendous convergence memory and calculation time. Besides, the scattering theory method has not yet been used to calculate the NFRHT between overlapping nanogratings.

In this study, based on our previous work that calculates the NFRHT between overlapping nanogratings [60], we verify the validity and accuracy of the effective approximation NFRHT method to calculate the NFRHT between a pair of SiO_2_ nanogratings with arbitrary relative arrangements. We calculate the heat fluxes between two nanogratings with different separation distances, grating periods, and lateral displacements. Compared with the EMT method, this effective approximation method considers the surface pattern effects of nanostructures and can more accurately predict the change in heat flux as a function of both the grating period and lateral displacement. Furthermore, by adjusting the height of nanogratings, we identify a heat flux behavior which trends first downward and then upward as the grating height increases, which verifies that the NFRHT between the side faces of the gratings greatly affects the NFRHT between a pair of nanogratings. Therefore, the effective approximation NFRHT method in this study provides a generalized and efficient path for calculating the NFRHT between nanogratings with arbitrary relative arrangements.

## 2. Theoretical Model and Method

A pair of one-dimensional (1D) movable SiO_2_ nanogratings is depicted schematically in Figure 1. The SiO_2_ nanograting period, depth, and ridge width are denoted as Λ = 100 nm, *h* = 100 nm, and *w* = 20 nm, respectively. The grating filling ratio is then *ϕ* = *w*/Λ = 0.2. Each nanograting has a 20 nm thick SiO_2_ layer below, followed by a 100 nm thick Au layer deposited on a substrate. The emitter (top nanograting) and receiver (bottom nanograting) temperatures are *T*_1_ = 301 K and *T*_2_ = 300 K, respectively. These temperature and structural properties are used as the default case in this work unless otherwise specified. The nanogratings face each other with a vacuum gap *D* and are shifted by a lateral displacement *a* or *b* as shown in Figure 1b. The distance *L* indicates the gap between the bottom thin film layers of the two nanogratings. 

The NFRHT between closely spaced objects can be calculated by [1,13,14,61,62]
(1)$$Q_{1\to 2}\left(T_{1},T_{2},L\right)\;=\int_{0}^{\infty }\frac{d\omega }{2\pi }\left[\Theta \left(\omega ,T_{1}\right)\;-\;\Theta \left(\omega ,T_{2}\right)\right]\int_{0}^{\infty }\frac{k_{\rho }{\mathrm{dk}}_{\rho }}{2\pi }\xi \left(\omega ,k_{\rho }\right),$$
where $\Theta \left(\omega ,T_{1}\right)\;=\;\left(\omega /2\right)\coth(\omega /2k_{B}T)$ is the energy of the harmonic oscillator. The function $\int_{0}^{\infty }\frac{k_{\rho }dk_{\rho }}{2\pi }\xi \left(\omega ,k_{\rho }\right)$ is the spectral transmissivity of the radiative transport between media 1 and 2 separated by distance L, where $\xi \left(\omega ,k_{\rho }\right)$ is the energy transmission coefficient.

A schematic of the NFRHT between a pair of movable SiO_2_ nanogratings is shown in Figure 2. The EMT method treats the nanograting structure as a homogeneous film, which makes it inaccurate for many geometrical cases such as nanogratings. For this reason, this work introduces an effective approximation NFRHT method which takes the geometric shape factors of nanostructures into account and employs it to calculate the NFRHT between two nanogratings. The effective approximation method calculates the total heat flux (*Q*_T_) between the nanograting features in the blue dashed box by adding up all the heat flux components between the two parallel subplates of the nanostructures. On one hand, when a pair of nanogratings face each other without a lateral displacement (Facing Case), *Q*_T_ is divided into four heat flux components (*Q*_1_~*Q*_4_), as shown in Figure 2a. *Q*_1_ is the near-field radiative heat flux between the top grating faces of the nanogratings. *Q*_3_ is the near-field radiative heat flux between the thin films under the nanograting. *Q*_2_ and *Q*_4_ are the near-field radiative heat fluxes between the side faces of nanogratings. On the other hand, when there is a lateral displacement between nanogratings (Offset Case), *Q*_T_ has six heat flux components (*Q*_1_~*Q*_6_), as shown in Figure 2b. *Q*_1_, *Q*_3_, *Q*_4_, and *Q*_5_ are longitudinal heat flux components, and *Q*_2_ and *Q*_6_ are the near-field radiative heat fluxes between the side faces of nanogratings. Here, we do not consider the near-field radiative heat fluxes between the side faces and horizontal faces of nanogratings because the well-known expression of NFRHT is applicable to parallel objects instead of vertical objects.

In order to calculate radiative heat fluxes between the left and right sides of the nanogratings, we introduce the geometric view factor *X*. The geometric view factor *X* in Facing Case and both *X_a_* and *X_b_* in Offset Case are expressed as
(2)$$X=\frac{\left(\sqrt{L^{2}+{\left(\Lambda -w\right)}^{2}}+\sqrt{{\left(2h-L\right)}^{2}+{\left(\Lambda -w\right)}^{2}}\right)-2\left(\sqrt{{\left(L-h\right)}^{2}+{\left(\Lambda -w\right)}^{2}}\right)}{2h},$$
(3)$$X_{a}=\frac{\left(\sqrt{L^{2}+a^{2}}+\sqrt{{\left(2h-L\right)}^{2}+a^{2}}\right)-2\left(\sqrt{{\left(L-h\right)}^{2}+a^{2}}\right)}{2h},$$
(4)$$X_{b}=\frac{\left(\sqrt{L^{2}+b^{2}}+\sqrt{{\left(2h-L\right)}^{2}+b^{2}}\right)-2\left(\sqrt{{\left(L-h\right)}^{2}+b^{2}}\right)}{2h}.$$

Then, the total heat flux *Q*_T_ in two cases ($Q_{T}^{1}$ in Facing Case and $Q_{T}^{2}$ in Offset Case) can be expressed as follows.
(5)$$Q_{T}^{1}=\sum_{i=1}^{4}Q_{i}=\;\left(T_{1}-T_{2}\right)\left(\frac{A_{1}}{\Lambda R_{1}}+\frac{A_{2}X}{\Lambda R_{2}}+\frac{A_{3}}{\Lambda R_{3}}+\frac{A_{4}X}{\Lambda R_{4}}\right),$$
(6)$$Q_{T}^{2}=\sum_{i=1}^{6}Q_{i}=\;\left(T_{1}-T_{2}\right)\left(\frac{A_{1}}{\Lambda R_{1}}+\frac{A_{2}X_{a}}{\Lambda R_{2}}+\frac{A_{3}}{\Lambda R_{3}}+\frac{A_{4}}{\Lambda R_{4}}+\frac{A_{5}}{\Lambda R_{5}}+\frac{A_{6}X_{b}}{\Lambda R_{6}}\right),$$
where *R_i_* represents the NFRHT thermal resistance between each plane. *A_i_* represents the corresponding area considered in the heat transfer calculation.

## 3. Results and Discussion

Figure 3 shows the heat flux *Q* in one period versus different vacuum gaps *D* (D≥Λ) in Facing Case calculated by our effective approximation NFRHT method and the EMT method, respectively. It is obvious that the total heat flux calculated by the effective approximation method (*Q*_T_) is larger than that calculated by the EMT method (*Q*_EMT_). However, these two results do have the same trend with an increase of gap *D*. Figure 3 also elucidates the distribution of four heat flux components with a change in *D*. The heat flux component *Q*_1_ is larger than *Q*_EMT_, while *Q*_2_, *Q*_3_, and *Q*_4_ are less than *Q*_EMT_. We identify that *Q*_T_ is larger than *Q*_EMT_ due to the near-field radiative heat fluxes between the top grating faces of the nanogratings (*Q*_1_), which make a significant contribution to *Q*_T_ due to the small separation of these subfaces. In other words, the effect of longitudinal nanograting movement on the radiative heat transfer between nanogratings is mainly attributed to the smaller distance between two parallel subplates of the nanogratings.

To prove the accuracy of the effective approximation method in considering the geometric shape factors of nanostructures, Figure 4a shows three kinds of nanograting structures with the same filling ratio schematically: Case 1, with Λ_1_ = 50 nm and *ϕ* = 0.2; Case 2, with Λ_2_ = 75 nm and *ϕ* = 0.2; and Case 3, with Λ_3_ = 100 nm and *ϕ* = 0.2. The vacuum gaps are kept constant at *D* = 120 nm for these three cases. Figure 4b displays the heat flux between each nanograting case in control volumes of equal sizes (represented by the blue dotted box), with three different grating periods (50 nm, 75 nm and 100 nm), calculated by the effective approximation NFRHT method and the EMT method, respectively. An increase in the nanograting period will weaken the NFRHT between the nanograting side faces because this modification increase extends the distance between the nanogratings’ side faces. It is apparent that the heat flux calculated by our method decreases with the increase of the grating period, while the heat flux calculated by the EMT method does not change at all. As the EMT method treats the grating structure as a homogeneous film, the grating period does not affect the heat flux between a pair of gratings with identical filling ratios. This means that the EMT cannot accurately analyze the effect of geometric shape factors on the heat flux between nanostructures. Conversely, the heat flux calculated by our method does change when the grating period is modified, indicating that the changes of the nanograting period are taken into account with this method. Therefore, compared with the EMT method, our effective approximation NFRHT method can accurately predict the radiative heat flux between nanogratings with various surface patterns.

In order to explain why the EMT-calculated heat flux does not change with the grating period, Figure 5 illustrates the refractive indices of SiO_2_ nanogratings with different grating periods and filling ratios. It is apparent that the real part (*n*) and imaginary part (κ) are not functions of the grating period, while the filling ratio has a great effect on the refractive indices of the SiO_2_ nanogratings. Therefore, the EMT method cannot accurately predict the change of heat flux caused by the effects of nanostructure surface patterns.

To further illustrate the advantage of our effective approximation method when calculating the NFRHT between a pair of movable nanogratings, Figure 6a shows the simulated heat flux *Q* against different lateral displacements of nanogratings using the effective approximation NFRHT method and the EMT method. A vacuum gap of *D* = 200 nm is considered within this calculation. It is obvious that the distribution of heat flux with varying lateral displacements calculated by the effective approximation method is variable and symmetrical, as can be expected. Meanwhile, when the lateral displacement is *a* = *b* = 30 nm, there is a minimum value of the heat flux. However, the heat flux calculated by the EMT method remains constant with varying lateral displacements. Therefore, compared with the EMT method, the effective approximation NFRHT method can indicate the effect of the lateral displacement on the heat flux between a pair of nanogratings. Figure 6b illustrates the distributions of six heat flux components with different lateral displacements. It is evident that both *Q*_1_ and *Q*_4_ remain the same due to the constant gap. The opposing trends of *Q*_3_ and *Q*_5_ with the lateral displacement are consistent with the changes of their corresponding NFRHT areas (i.e., *A*_3_ and *A*_5_). Similarly, the opposing trends of *Q*_2_ and *Q*_6_ with lateral displacement are due to the changes in the respective lateral distances between the side faces of the gratings. By distinguishing these six heat flux components, it is apparent that the symmetrical distribution of heat flux with the lateral displacements are mainly due to the contribution of the NFRHT between the sides of the nanogratings (*Q*_2_ and *Q*_6_). By considering both Figure 6a,b, we can identify why the NFRHT calculated by the effective approximation method is larger than that of the EMT. The effective approximation NFRHT method considers the NFRHT between the side faces of gratings with changes in lateral displacement, which accounts for the majority of the total heat flux, while the EMT does not consider these effects. As a result, our effective approximation NFRHT method is more apt to consider the NFRHT between two nanogratings.

In order to further investigate the importance of the NFRHT between the side faces of the gratings, Figure 7 shows the heat flux between a pair of nanogratings with various grating heights *h* calculated by the effective approximation NFRHT method and the EMT method. The vacuum gap is kept constant at *D* = 200 nm for this case. When the grating height *h* is less than the 20 nm, the heat fluxes calculated by both methods decrease with increases of the grating height *h*. Based on the effective approximation method, this is attributed to the increase of the NFRHT heat flux components between the sides of the nanogratings (*Q*_2_ and *Q*_4_), which are less than the decrease of heat flux component *Q*_3_. When the grating height *h* continues to increase (*h* > 20 nm), the effective approximation NFRHT-calculated flux trends upwards, opposite to that of the EMT method. In this effective approximation method, the heat flux components *Q*_2_ and *Q*_4_ increase greatly with grating height once *h* exceeds 20 nm due to the increase of the corresponding heat transfer areas (*A*_2_ and *A*_4_) and the geometric view factor *X*, both of which cause the increase of *Q*_2_ and *Q*_4_ to exceed the decrease of *Q*_3_. Conversely, the EMT method only accounts for the decrease in the longitudinal heat flux due to the additional grating height and does not consider the NFRHT interaction between the two side faces of the nanogratings (*Q*_2_ and *Q*_4_). As a result, the calculated flux of the EMT method continues to trend down as the grating height is increased. Thus, the NFRHT between the side faces of the gratings plays a very important role in the total NFRHT between a pair of nanogratings. Meanwhile, when the grating height *h* is less than 10 nm, the radiative heat fluxes calculated by both methods are very close, as shown in the inset. After further calculation, when the distance is maintained at 150 nm or 250 nm, the trends of heat flux calculated by the two methods are similar to the aforementioned trends, respectively. This also proves that the EMT method is less accurate for this geometry as it treats the nanograting as the homogeneous film and does not consider the NFRHT between the side faces of gratings. Therefore, we assert that our effective approximation NFRHT method can accurately predict the heat flux between nanogratings with various structures at different values of relative displacement.

## 4. Conclusions

In summary, we proposed and verified an effective approximation method to study the NFRHT between a pair of movable nanoscale gratings. Compared with the EMT method, our effective approximation NFRHT method takes the geometric shape factors of nanostructures into account precisely, enabling this method to accurately predict the changes in the heat flux between two nanogratings with different separation distances, grating periods, and lateral displacements. Meanwhile, we also proved that the grating period and the lateral displacement have a great effect on the modulation of NFRHT between two nanogratings via the effective approximation NFRHT method. Additionally, by increasing the height of the nanogratings, it is possible to realize opposing heat flux trends when using our effective approximation method, which verifies that the NFRHT between the side faces of gratings greatly affects the total NFRHT between a pair of nanogratings. This work offers a generalized and efficient numerical calculation method to calculate the NFRHT between nanostructures with arbitrary relative arrangements. The effective approximation NFRHT method will shed significant light on applications in contactless thermal management, near-field energy harvesting, conversion and storage, and relieving adhesion problems of thermoelectric devices.

## Figures and Tables

**Figure 1 materials-15-00998-f001:**
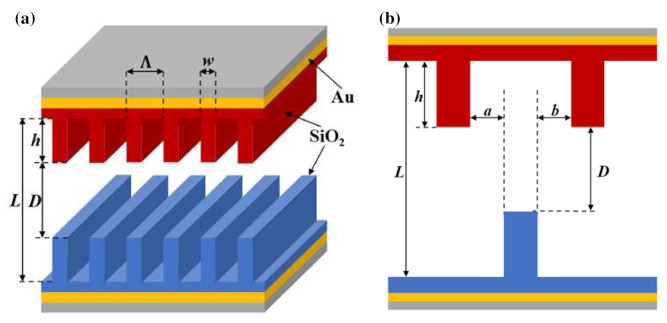
(**a**) Schematic of a pair of movable SiO_2_ nanogratings. The SiO_2_ nanograting period, depth, and ridge width are Λ = 100 nm, *h* = 100 nm, and *w* = 20 nm, respectively. The two SiO_2_ nanogratings face each other with a vacuum gap *D* and the distance *L* indicates the distance between the bottom thin film layers of the nanogratings. (**b**) A sketch of two 1D SiO_2_ nanogratings with a lateral displacement *a* and *b*.

**Figure 2 materials-15-00998-f002:**
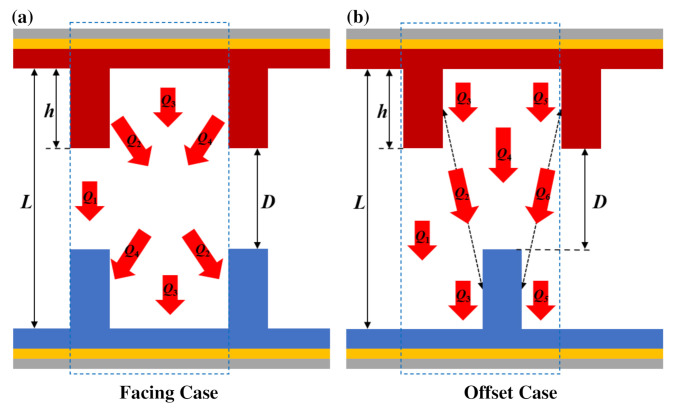
Schematic of NFRHT between a pair of movable nanogratings. The total heat flux (*Q*_T_) between the nanogratings shown in the blue dotted box is divided into (**a**) four heat flux components in Facing Case and (**b**) six heat flux components in Offset Case.

**Figure 3 materials-15-00998-f003:**
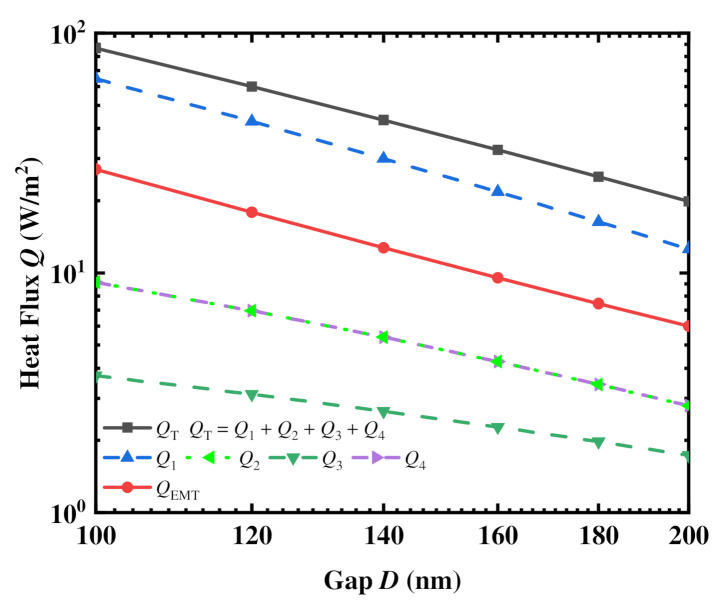
Comparison of radiative heat fluxes between the nanogratings in Facing Case calculated by both the effective approximation NFRHT method and the EMT method for a vacuum gap *D* larger than the nanograting period (*D* > 100 nm).

**Figure 4 materials-15-00998-f004:**
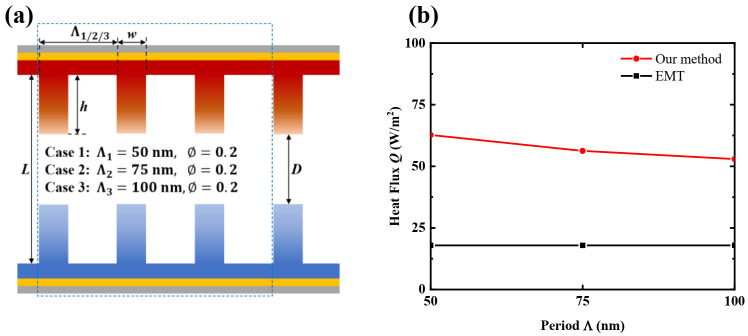
Schematic illustrations of three nanograting structures with identical filling ratios: (**a**) Case 1, Λ_1_ = 50 nm and *ϕ* = 0.2; Case 2, Λ_2_ = 75 nm and *ϕ* = 0.2; and Case 3, Λ_3_ = 100 nm and *ϕ* = 0.2. (**b**) Comparison of the heat flux between the nanogratings in equally sized control volumes (represented by the blue dotted boxes) calculated by the effective approximation NFRHT method and the EMT method for all three cases.

**Figure 5 materials-15-00998-f005:**
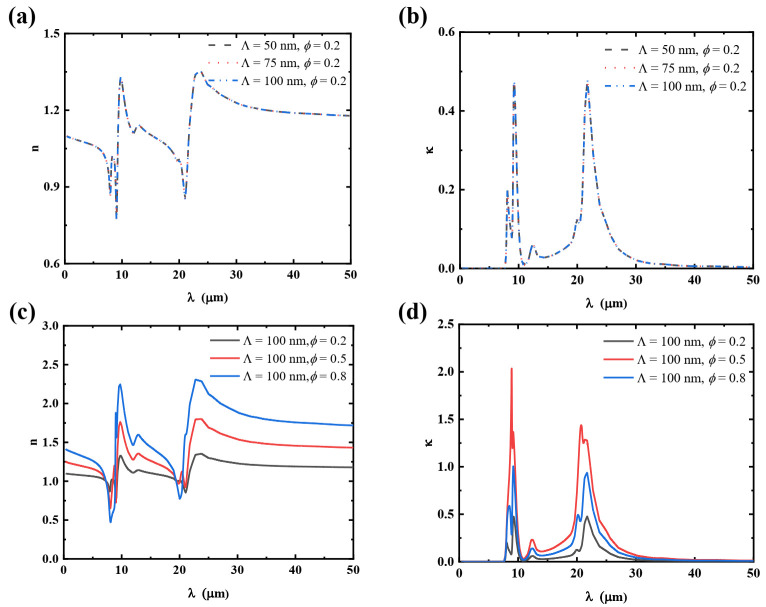
Refractive indices of SiO_2_ nanogratings with different grating periods and filling ratios. The comparison of real part (*n*) and imaginary part (κ) of refractive index with **(a**,**b**) constant filling ratios and different grating periods, and (**c**,**d**) with constant grating periods and different filling ratios.

**Figure 6 materials-15-00998-f006:**
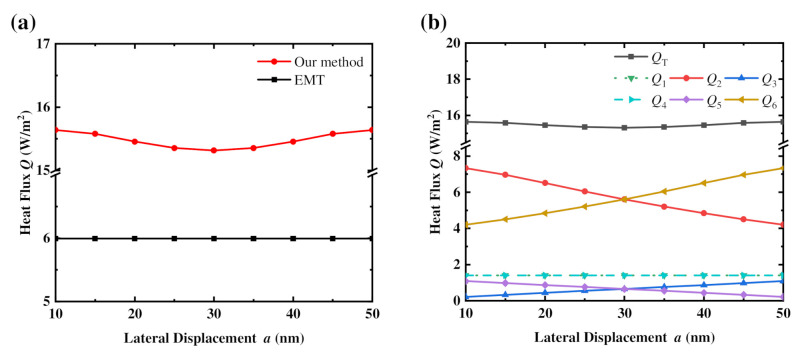
(**a**) Radiative heat fluxes for different lateral displacements *a* of nanogratings with a gap *D* = 200 nm calculated by the effective approximation NFRHT method and the EMT method. (**b**) Distributions of six heat flux components with different lateral displacements *a*.

**Figure 7 materials-15-00998-f007:**
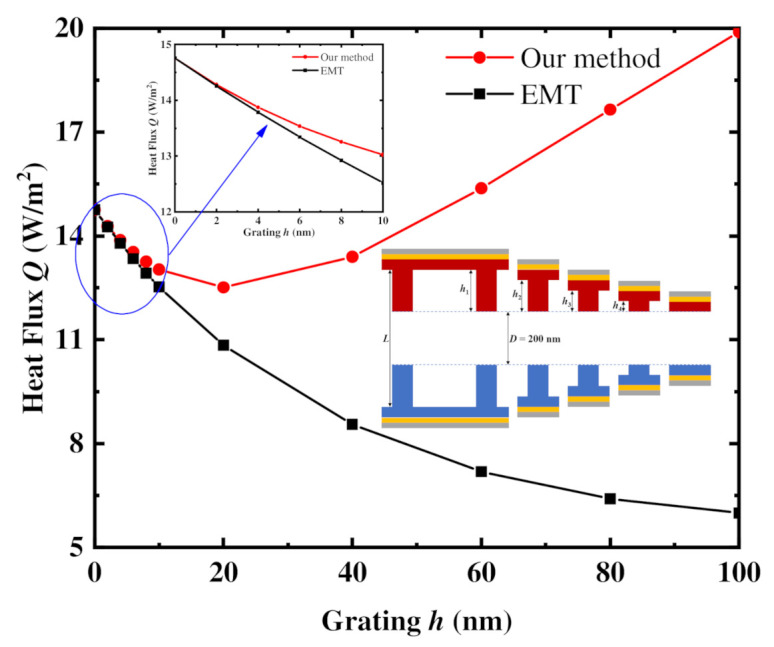
Comparison of heat fluxes between the nanogratings with a vacuum gap *D* = 200 nm for different grating heights *h* calculated by the effective approximation NFRHT method and the EMT method.

## Data Availability

The data that support the findings of this study are available from the corresponding author upon reasonable request.

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
