# Peer review of "Effective Approximation Method for Nanogratings-induced Near-Field Radiative Heat Transfer"

_materials, 2022, doi:10.3390/ma15030998_

Round 1
Reviewer 1 Report
In this study, the heat transfer enhancement provided by the near field radiation between the silica nano-gratings is studied. One kind of effective approximation method is employed by the author, and the influence of the geometric parameters such as the lateral distance, grating heights, and the separation distance are estimated by author. The advantages of this approximation method are declared by comparison the results of effective medium method (EMT) and the effective approximation method, but with no solid supports like experiments results, which is widely carried out by the former studies. Only the calculation results seem to be not enough to show the innovation of the article and fulfill the journal's requirements for research articles.
The following are specific suggestions:
- In the section of Introduction, the author introduces former studies about the near field radiation, without the introduction on the calculation method, which should be done to show the advantages of the approximation method employed by author, and how this method overcomes the shortcomings of other methods.
- In the section of theoretical model and method, the author gives the relationship used to calculate the heat flux between two closely spaced bodies by the equation (1). This equation is widely used, and not firstly proposed by the reference 13,14. The author should reconfirm the references.
- In the section of theoretical model and method, the author takes the influences raised by the nano-gratings’ special geometry, with the employment of the geometric view factor theory. This theory is mostly used in the traditional radiation calculation, and whether it is appropriate to be used in the near field radiation should be further confirmed and clarified, since the near field radiation theory is based on the fluctuation dissipation.
- In the section of result and discussion, the author shows the different results achieved by the EMT and the approximation method. Since, the EMT results didn’t show any variations between different grating period, so the author comes to the conclusion that the approximation method can achieve more accurate results than the EMT method. However, to calculate the heat flux provided by the near-field radiation, there are also many other methods. Specific to the grating geometry there are also some mature paths.
- The whole manuscript is based on the theoretical calculation, not enough to support a research article. And the results achieved by the calculation can’t certify the advantages of the method and the innovation of this article. So, the experiment data should be added to give further research, and give solid proof. Also, if the geometric parameters of the grating are optimized on the aim of energy management, more discussions are needed to supplement.
Reviewer 2 Report
In this paper, the authors analytically demonstrate an effective approximation method to study the near-field radiative transfer between a pair of movable nanoscale gratings. Their method allows them to precisely account for the geometric shapes of the nanostructures and hence model precisely the changes in heat flux between two nanogratings. In this study, the authors also assess the effects of dimensions that on the modulation of NFRHT, with aim for tuning the NFRHT between the nanogratings. This paper is well-written and a few minor comments to improve it are as follows:
- It is unclear how much of the nanogratings protrude outside the layer of SiO2, for example, the authors demonstrate a nanograting of height h, but how much of this is left on the Au surface?
- In addition, would the thickness of the remaining SiO2 layer matter in the heat flux, perhaps you can comment on this on the paper.
- Does your method assume the length of the grating is infinite, and how long should the gratings be for this method to remain valid?
- It is not useful to have 3 figures in a panel representing similar things such as Figure 4, where you show identical filling ratios, which is obvious. Perhaps you can consider showing only two of them.
- With longer grating heights, the approximation method diverge significantly from the EMT method, and indicating that with a height of around 100 nm could have more heat flux than between flat surfaces. This could be because, as explained by the authors, the opposite sides of the walls begin to dominate. However, does the point where the divergence occurs change with the gap distance D? Perhaps a few more plots for Figure 7 with different D would add more substance to your claim.
- How would a material with different thermal properties change the results, and is the method valid as well. Perhaps, illustrating this with silicon nitride or poly-silicon could demonstrate the robustness of your method.
Reviewer 3 Report
Authors introduce the effective approximation NFRHT method that considers the effects of surface patterns on the NFRHT. Meanwhile, we calculate the heat flux between a pair of silica (SiO2) nanogratings with various separation distances, lateral displacements, and grating heights with respect to one another. Numerical calculations show that, when compared with the EMT method, here the effective approximation method is more suitable for analyzing the NFRHT between a pair of relatively displaced nanogratings.
First of all, references citation is poor, all of them being weekly discussed. So, state of the art is incomplete and needs refinement and a better approach.
Results are not properly discussed and no comparison with state of the art is noticed.
Conclusion does not reflect the content.
What are the applications? What are authors contribution towards state of the art improvement?
These are relevant basic questions that authors did not addressed in their paper.
Reviewer 4 Report
The topic is interesting and the manuscript is organized with figures. The topic falls in the journal scope.
However, the reviewer has some believes a revision is needed to address the following concerns before it can be accepted for publication.
- Components of the proposed element have a small thickness. Hence, in order to enrich the Introduction in the framework of mechanics of small-scale structures, some recent relevant papers on nonlocal models can be considered such as
On thermomechanics of multilayered beams (2020) International Journal of Engineering Science, 155, art. no. 103364;
Effects of nonlocal thermoelasticity on nanoscale beam sbased on couple stress theory; Math Meth Appl Sci. 2020;1–17;
Axial and torsional free vibrations of elastic nano-beams by stress-driven two-phase elasticity (2019) Journal of Applied and Computational Mechanics, 5 (2), pp. 402-413.
- Line 95 - The assumption that when a pair of nanogratings face each other without a lateral displacement, Q_T is divided into four heat flux components should be better enlightened.
- Eqs. (2) and (3) – These Eqs. should be clarified from a mechanical point of view.
- Novelties of the manuscript should be better pointed out.
Reviewer 5 Report
In the manuscript, the authors reported an approach toward 'Effective Approximation Method for Nanogratings Induced Near-Field Radiative Heat Transfer'. Generally current work is well carried out but the authors should try to emphasize better the importance of current manuscript in order to attract the readership of this journal. Besides, the publication of the work in this journal can be justified after the authors consider the following major points.
Comments:
1. The authors should try to give details about the preparation of the materials.
2. The authors should explain the importance of the work in detail in order to attract the readership of this journal.
3. The authors should try to give advantegous of using of the prepared materials compared to the commercial and/or others in literature in terms of cost, efficiency, figures of merit etc.
4. The introduction part is too casual. The authors should try to give some recent references such as Environmental Chemistry Letters 19, 2185-2207 (2021); Topics in Current Chemistry 2020, 378 (6), 1-43; Environmental Chemistry Letters 19, 375-439 (2021).
5. Manuscripts published in this journal must explain the significant advances provided in approaches and understanding compared to previous literature, and/or demonstrate convincingly potential in new applications. The Conclusions of your paper are especially important for this. Therefore, please try to sharpen this further. The optimal Conclusion should include:
* A summary of your findings.
* A synopsis of your new concepts and innovations.
* A brief restatement of your hypotheses.
* Your vision for future work.
6. Language needs substantial improvement.
7. There are many typos and grammatical issues.
Round 2
Reviewer 1 Report
After revision, the author made some changes and improvements in accordance with the review commends, but there are still some questions that the author did not give a detailed answer or give a further in-depth discussion. Among these questions, some determine the reliability of this article and whether it can be accepted as a qualified research paper.
The following are specific suggestions:
- In the section of Introduction, after the revision, the author added the introduction on calculation methods to calculate the near-field radiation heat transfer, but still not showing the advantages of the approximation method employed by author. The disadvantage of the former method seems to be long calculation time and tremendous convergence memory, but the author didn’t show the improvement on these two important points.
- In the section of theoretical model and method, the author takes the influences raised by the nano-gratings’ special geometry, with the employment of the geometric view factor theory. After the revision, the author still not gave the reason why it is appropriate to use geometric view factor theory in the near field radiation. It’s necessary to give provement on this, since the following calculation is totally based on this.
- In the section of result and discussion, the author still not gave the comparison to the results calculated by the article method and other mature paths. Actually, there are many paths to calculate near-field radiation between grating geometry, the comparison to EMT method is far from enough, and the advantage of the author’s method still can’t be clarified.
- After the revision, the whole manuscript is still based on the theoretical calculation, not enough to support a research article. And the results achieved by the calculation can’t certify the advantages of the method and the innovation of this article. So, the experiment data should be added to give further research, and give solid proof.
Reviewer 3 Report
The article was revised, but this reviewer cannot see the relevance of the research, if compared with state of the art. There is a parametric study and I cannot recommend it to be published due to its poor approach and results discussion.
Reviewer 5 Report
Accept as it is
Author Response
Thank you very much for your comments. We are very grateful for the time you spent reviewing this manuscript.